# Un enfoque matheurístico para el problema de la $p$-mediana inducida con mejora

**Sergio Salazar**
Dpto. de Informática y Estadística
Universidad Rey Juan Carlos
Madrid, Móstoles, 28933
sergio.salazar@urjc.es

**J. Manuel Colmenar**
Dpto. de Informática y Estadística
Universidad Rey Juan Carlos
Madrid, Móstoles, 28933
josemanuel.colmenar@urjc.es

## Abstract

Los problemas de ubicación de instalaciones (FLPs por sus siglas en inglés) son una familia de problemas de optimización con gran impacto social. En concreto, el problema de la $p$-mediana inducida con mejora (*Induced p-median Problem with Upgrading*, IpMU) es una variación del problema clásico de la $p$-mediana, donde se separan los conceptos de coste de transporte y distancia en dos grafos distintos. Además, se añaden un conjunto de variables de decisión para relajar el grafo de costes, de manera que las aristas entre nodos pueden ser reducidas para mejorar las rutas asociadas entre las medianas elegidas y los clientes. En este trabajo se propone un algoritmo matheurístico donde se establece un esquema de resolución en dos fases, estudiando el problema de medianas y el problema de mejora de manera independiente. En esta aproximación se obtienen resultados prometedores en comparación con el estado del arte, basado completamente en modelos matemáticos, pero con amplio margen de mejora para el trabajo futuro.

## 1 Introducción

La familia de problemas relacionados con la ubicación de instalaciones, conocidos en inglés como *Facility Location Problems* (FLPs), ha despertado un gran interés en la comunidad de investigación operativa. Estos problemas se centran en determinar la ubicación óptima de un conjunto de elementos en diversos entornos, como espacios industriales, redes de infraestructura físicas como carreteras, e incluso redes informáticas [1] [2].

Los problemas de ubicación de instalaciones se clasifican según el dominio del problema. En su vertiente continua no existe un conjunto candidato para las ubicaciones, sino que se considera una zona acotada del plano. En este grupo se pueden encontrar algunos problemas clásicos, como el Problema de Weber [3], y aproximaciones más recientes, como el problema de ubicación de múltiples instalaciones desagradables (*Multiple Obnoxious Facilities Location Problem*, MOFLP) [4] [5].

En su variante combinatoria, los problemas de ubicación identifican un conjunto de candidatos donde colocar las instalaciones. Normalmente, este conjunto se define como un subconjunto de nodos en un grafo, donde las aristas representan comunicaciones, que pueden estar ponderadas atendiendo a diferentes métricas como las distancias o los costes. Un ejemplo clásico es el problema de la $p$-mediana [6], que busca seleccionar $p$ nodos de forma que se minimice la distancia del resto al conjunto de medianas. Por otro lado, existen variantes alternativas como el problema de la $p$-mediana desagradable (*obnoxious p-median*) donde se busca maximizar la mínima distancia de cada nodo al conjunto de medianas [7], o el problema de la $p$-mediana ponderada (*weighted p-median*) donde, además de seleccionar la localización, se debe cumplir la demanda de los nodos clientes [8].

XVI XVI Congreso Español de Metaheurísticas, Algoritmos Evolutivos y Bioinspirados (maeb 2025).

Entre estas variantes combinatorias destaca el problema de la *p*-mediana inducida con mejora (*Induced p-Median Problem With Upgradings*, IpMU) [9]. Esta variante del problema de la *p*-mediana busca minimizar el coste de comunicar el conjunto de nodos del grafo con el conjunto de medianas seleccionadas. Sin embargo, separa los conceptos de tiempo de transporte (distancia) y coste; de esa manera, las aristas del grafo de la instancia estarán ponderadas por dos valores, representando ambas métricas. Los nodos serán asignados según el tiempo de transporte, tomando siempre el camino más corto del conjunto de medianas al nodo, pero el coste será calculado mediante el segundo valor. Esta versión del problema se acerca más a un escenario real donde los valores de tiempo de transporte (distancia) y coste no tienen por qué estar siempre relacionados. Además de esta característica, el problema añade una variable de presupuesto que permite relajar los valores de coste asociados a las aristas, compensando el coste. De esta manera, se pueden obtener mejores valores en la función objetivo del problema [9].

Para la resolución de este problema se ha optado por el diseño de un algoritmo matheurístico que combina la aplicación de modelos matemáticos con una metodología metaheurística basada en el procedimiento de búsqueda adaptativo aleatorizado y voraz (*Greedy Randomized Adaptative Search Procedure*, GRASP)[10]. En este trabajo se opta por una resolución en dos fases donde se separan el problema de ubicación del problema de relajación de aristas. En la primera fase se utilizará un algoritmo GRASP para la selección de medianas del problema, mientras que en la segunda fase se utiliza un modelo matemático para obtener la relajación óptima del grafo. En la comparación con el estado del arte, la propuesta matheurística obtiene todos los valores óptimos salvo uno, empleando un 33% menos de tiempo en las instancias más complejas.

El resto del artículo se organiza de la siguiente manera: en la Sección 2 se describe formalmente la modelización del problema; en la Sección 3 se desarrolla la propuesta de este trabajo para la resolución del problema IpMU; en la Sección 4 se hace una descripción del estado del arte y de la experimentación realizada, así como una comparativa sobre las instancias estudiadas; finalmente, en la Sección 5 se describen las conclusiones del trabajo y se apuntan vías de mejora.

## 2 Definición formal del problema

Para mayor completitud del artículo, se propone en esta sección una formalización detallada del problema a partir de la propuesta del estado del arte [9]. Sea $G = (V, A)$ un grafo dirigido, se considera que todos los nodos $V = \{1, ..., n\}$ representan clientes del problema. Cada cliente $i \in V$ tiene asignada una demanda $\omega_i \geq 0$. Sean $c^1$ y $c^2$ el conjunto de costes asociado al conjunto de aristas del grafo, para cierta arista $a \in A$ se denomina $c_a^1 \geq 0$ al tiempo de transporte por $a$ y a $c_a^2 \geq 0$ el coste de transporte por unidad de $a$.

Siguiendo la modelización original, se define el conjunto de nodos $V$ como el conjunto de candidatos para ser seleccionados como medianas, de las que deben ser escogidas $1 \leq p < n$ de ellas. Una vez el conjunto $S = \{s_1, ..., s_p\}$ de medianas es establecido, cada usuario debe ser servido por la mediana con menor tiempo de transporte. Más formalmente, sea $FP(i, j) \subset A$ el camino más corto desde el nodo $j \in V$ al nodo $i \in V$ según la medida $c^1$, la longitud de $FP(i, j)$ se define como $C_{i,j}^1$. Por tanto, cada cliente $i \in V$ será servido por la mediana $s \in S$ tal que $C_{s,i}^1 \leq C_{s',i}^1 \ \forall s' \in S$. Además se define el coste de transporte por unidad $C_{i,j}^2$ entre $i \in V$ y $j \in V$ como la suma de los costes $c^2$ asociados a $FP(i, j)$, es decir, $C_{i,j}^2 = \sum_{a \in FP(i,j)} c_a^2$.

Finalmente, se define el presupuesto $B > 0$ como el valor que puede ser utilizado para reducir el coste de transporte asociado a las aristas. La relajación para cada arco $a \in A$ del grafo está limitada por $u_a$.

El IpMU consiste en identificar $p$ medianas y distribuir el presupuesto $B$ entre los arcos del grafo de forma que se minimice la suma del coste de transporte de las medianas a sus correspondientes clientes, ponderados por la demanda solicitada.

La formulación del problema se muestra en la Ecuación (1), para la que se necesitan definir las siguientes variables:

- $x_{jj} \in \{0, 1\}$: Si el nodo $j$ es elegido como mediana.
- $x_{ij} \in \{0, 1\}$: Si el nodo $j$ es la mediana más cercana de $i$.

- $b_a \in [0, u_a]$ : La reducción del coste de transporte aplicada a la arista $a \in A$

$$
\min \quad \sum_{i \in V} \omega_i \sum_{j \in V} (C_{i,j}^2 - \sum_{a \in FP(i,j)} b_a) x_{ij}
$$

$$
\text{s.t.}
$$

$$
\begin{aligned}
& x_{ij} \leq x_{jj}, && \forall i \neq j \in V, \\
& \sum_{j \in V} x_{ij} = 1, && \forall i \in V, \\
& \sum_{j \in V} x_{jj} = p, && \\
& x_{jj} + \sum_{\substack{s \in V: \\ C_{is}^1 > C_{ij}^1}} x_{is} \leq 1, && \forall i,j \in V, \\
& \sum_{a \in A} b_a \leq B, && \\
& b_a \leq u_a, && \forall a \in A \\
& b_a \geq 0, && \forall a \in A \\
& x_{ij} \in \{0,1\}, && \forall i,j \in V.
\end{aligned}
\tag{1}
$$

En la Figura 1 se presenta una instancia de ejemplo para el IpMU donde se muestran los dos valores de cada arista duplicando el grafo. En este caso, se considera un presupuesto $B = 2$, una capacidad máxima de mejora para cada arista dada por $u_a = c_a^2$ y una demanda uniforme $\omega_i = 1$ para todos los clientes. La cantidad de medianas a colocar es $p = 2$. Aunque el modelo supone que los nodos del grafo están numerados de 1 a $n$, para facilitar la comprensión del ejemplo, se han etiquetado con las letras de la $A$ a la $E$.

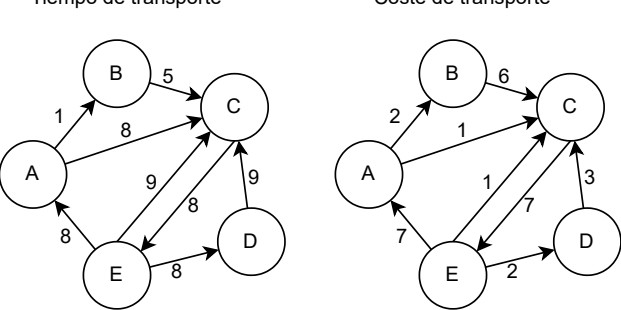

Figure 1: Ejemplo instancia IpMU. Se asume un presupuesto de $B = 2$, una capacidad de mejora $u_a = c_a^2$, una demanda $\omega_i = 1$ y una cantidad de medianas a colocar $p = 2$.

En este caso, la solución óptima para esta instancia, ilustrada en la Figura 2, consiste en ubicar las medianas en los nodos $A$ y $E$. El nodo $A$ atenderá a los nodos $B$ y $C$, ya que es su mediana más cercana. Por otro lado, el nodo $D$ será atendido por la mediana $E$ mediante la arista que los conecta. En cuanto a la asignación de presupuesto de mejora, se destinan las dos unidades a reducir el coste de la arista $A - B$, aprovechando que esta aparece en los caminos $FP(A, B)$ y $FP(A, C)$. El valor final de la función objetivo, calculada según el coste de transporte, es de 8 unidades: 6 unidades en el camino $FP(A, C)$, 0 unidades en el camino $FP(A, B)$ y 2 unidades en el camino $FP(E, D)$.

## 3   Propuesta algorítmica

El enfoque seguido por el estado del arte es la linealización del modelo observado en la Ecuación (1). En ese trabajo se proponen 3 modelos lineales distintos, de los cuales uno de ellos (*FL1* en [9]) obtiene los mejores resultados en tiempo de ejecución.

La propuesta de este trabajo es un algoritmo matheurístico [11]. Estos algoritmos están caracterizados por aunar una propuesta metaheurística con modelos exactos. En este caso, el esquema principal del algoritmo está basado en la metodología GRASP [10] donde se combina una construcción voraz aleatorizada con una fase de mejora en un esquema multiarranque. En cada construcción del algoritmo

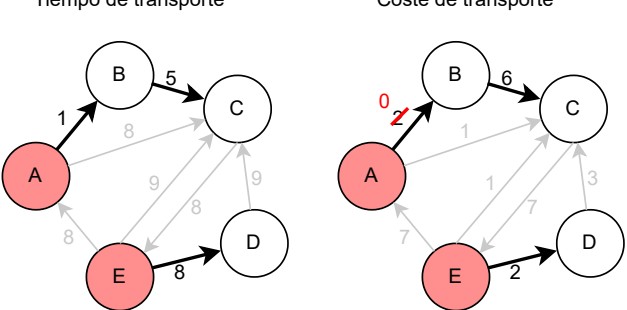

Figure 2: Representación de la solución óptima para la instancia de la Figura 1.

GRASP se seleccionarán las medianas que formarán parte de una solución. Para su evaluación, es necesario identificar las aristas del grafo de costes que serán relajadas. Este conjunto de aristas se determina mediante la resolución de un modelo matemático, que obtiene el conjunto óptimo de aristas relajadas en función de las medianas escogidas.

### 3.1   Resolución del subproblema de relajación de aristas.

En primer lugar, se define la solución del subproblema de relajación, ya que este se incorpora en el cuerpo del algoritmo de resolución del IpMU. En distintas fases del algoritmo se partirá de una solución total o parcial, que consiste en un conjunto de nodos seleccionados como medianas. Con este conjunto se ejecutará un modelo que obtiene la relajación del grafo de costes que minimiza la función objetivo.

Sea $S' = \{s_1, ..., s_q\}$ el conjunto de medianas seleccionadas, tal que $q \leq p$. En el caso de que se deba evaluar una solución parcial para el problema ($q < p$) su calidad vendrá definida por la calidad de esa solución en el problema restringido a $q$ medianas. En esta situación, véase que siguiendo la formulación del modelo en la Ecuación (1), las variables $x_{jj}$ y $x_{ij}$ toman el valor correspondiente según las medianas seleccionadas, mientras que aún falta por asignar las variables $b_a$.

En estas condiciones, los valores $FP(i, j), C_{ij}^1$ y $C_{ij}^2$ son constantes, ya que únicamente dependen de la selección de medianas. Siguiendo esta idea, se define el modelo del problema IpMU supeditado a $S'$ en la Ecuación (2). La solución de este modelo utiliza los valores anteriormente mencionados, que son constantes en la instancia, y la eliminación de restricciones no relacionadas con $b_a$. Nótese que el modelo de la Ecuación (2) no depende en ningún caso del valor $p$ por lo que se puede obtener la relajación para una solución completa o parcial.

$$\max \quad \sum_{i \in V} \sum_{\substack{j \in V : \\ x_{ij}=1}} \sum_{a \in FP(i,j)} \omega_i b_a$$

$$\text{s.t.}$$

$$\sum_{a \in A} b_a \leq B,$$
$$b_a \leq u_a, \qquad \forall a \in A$$
$$b_a \geq 0, \qquad \forall a \in A$$

(2)

El modelo propuesto consiste en un problema lineal con $|A|$ variables y $2|A| + 1$ restricciones, lo que implica una gran simplificación respecto al modelo inicial. Para cada construcción de una solución en las distintas fases del algoritmo se lanzará la resolución del modelo, obteniendo así la relajación del grafo de costes y permitiendo el cálculo de la función objetivo (denotada como $\mathcal{F}$) para la solución total o parcial estudiada ($S$), siguiendo el esquema que se muestra en el Algoritmo 1. En el paso 1 del algoritmo se inicializa la matriz $x$ de asignaciones a 0. Posteriormente, por cada mediana $j$ escogida se establece la variable $x_{jj}$ a 1 y se imponen las variables $x_{ij}$ de forma que la mediana $j$ sea la más cercana al nodo $i$ (pasos 2 a 5). En el paso 6 se establece el vector de valores reales $b$, donde $b_a$ representa la relajación impuesta a la arista $a$ del grafo de costes. Este valor se obtiene ejecutando el modelo de la Ecuación (2), representado por la función *relaxEdges*. Finalmente, una vez todas las variables han sido establecidas, se devuelve el valor de la función objetivo.

**Algorithm 1:** $\mathcal{F}(S)$

1: $x \leftarrow \{0\}_{n \times n}$
2: $x_{jj} \leftarrow 1 \ \forall j \in S$
3: **for** $j \in S$ **do**
4:     $x_{ij} \leftarrow 1 \ \forall i \in \{i \in V : C^1_{i,j} \leq C^1_{i,k} \ \forall k \in S\}$
5: **end for**
6: $b \leftarrow relaxEdges(x)$
7: **return** $\sum\limits_{i \in V} \omega_i \sum\limits_{j \in V} (C^2_{i,j} - \sum\limits_{a \in FP(i,j)} b_a) x_{ij}$

Véase que la propuesta de separar el conjunto de aristas, a través de una resolución óptima dado el conjunto $S$, aligera en gran medida la combinatoria con la que el algoritmo metaheurístico debe trabajar. De este modo, únicamente deberá escoger $p$ medianas sobre un total de $n$ nodos, es decir, hay un total de $\binom{n}{p}$ soluciones, que aunque sigue un comportamiento factorial, es mucho menor que incluir las aristas del grafo que toma valores continuos, ya que $b_a$ es un valor no discreto.

## 3.2 Algoritmo para la selección de medianas

La propuesta para la resolución del IpMU está basada en un algoritmo GRASP [10]. Este algoritmo consta de una fase de construcción aleatorizada y una fase de mejora que ha sido implementada a través de una búsqueda local.

### Fase constructiva

El Algoritmo 2 muestra el pseudocódigo de selección de medianas para el problema IpMU. En el primer paso se inicializa una solución vacía, a la que se irán añadiendo las medianas elegidas por el algoritmo. Seguidamente, en el paso 2, se inicializa la lista de candidatos ($CL$) que incluye todos los nodos del grafo. Posteriormente se comienza el bucle principal del algoritmo. Se identifican $f_{max}$ y $f_{min}$ como los valores máximo y mínimo tras evaluar la inclusión de todos los nodos candidatos a la solución $S$ (pasos 4 y 5). Con estos valores y el parámetro $\alpha$, recibido como entrada, se establece un valor umbral $th$ (paso 6) para construir la lista de candidatos restringida ($RCL$), que incluye aquellos nodos cuya evaluación está por debajo del umbral $th$ al añadirse a la solución (paso 7). Entre estos elementos se escoge uno de manera aleatoria, se añade a la solución y se elimina de $CL$ (pasos 8 hasta 10). Una vez la solución alcanza $p$ medianas escogidas, el bucle principal termina y se devuelve la solución.

**Algorithm 2:** $GRASPBuild(\alpha, V)$

1: $S \leftarrow \emptyset$
2: $CL \leftarrow V$
3: **while** $|S| < p$ **do**
4:     $f_{max} \leftarrow \max\limits_{i \in CL} \mathcal{F}(S \cup \{i\})$
5:     $f_{min} \leftarrow \min\limits_{i \in CL} \mathcal{F}(S \cup \{i\})$
6:     $th \leftarrow f_{max} + \alpha \cdot (f_{min} - f_{max})$
7:     $RCL \leftarrow \{i \in CL : \mathcal{F}(S \cup \{i\}) \leq th\}$
8:     $s \leftarrow random(RCL)$
9:     $S \leftarrow S \cup \{s\}$
10:    $CL \leftarrow CL \setminus \{s\}$
11: **end while**
12: **return** $S$

**Fase de mejora**

La fase de mejora del algoritmo GRASP está implementada a través de una búsqueda local definida por un movimiento de intercambio en el conjunto de medianas. Este movimiento consiste en eliminar un nodo $i \in S$ e introducir un nuevo nodo $j \in V \setminus S$.

En el Algoritmo 3 se muestra el pseudocódigo de este procedimiento. Véase que la estrategia de búsqueda implementada es *Best Improvement* [12], es decir, se explora todo el vecindario en busca del mejor movimiento posible. En el primer paso se establece una solución $S_{best}$ que almacenará la mejor solución encontrada por el algoritmo. Seguidamente se inicia un bucle que finalizará al encontrar un mínimo local. En cada iteración del bucle se busca el movimiento que suponga la mayor mejora de calidad de la solución actual, que será almacenada en $S'$, inicializada, a su vez, en el paso 5. Para identificar el movimiento se itera entre las medianas y los nodos clientes, construyendo una solución candidata resultado del intercambio de ambos elementos (pasos 6 hasta 8). En el paso 9 se comprueba si la nueva solución es mejor, en cuyo caso se almacena en $S'$, es decir, se ha encontrado una mejora en el vecindario actual. En ese caso, la nueva solución se almacena en $S_{best}$ y se cambia la variable $mejora$ para repetir la búsqueda en el vecindario de la nueva solución (pasos 14 hasta 17). Finalmente, se devuelve la mejor solución encontrada. Este método se puede adaptar de manera sencilla a la estrategia *First Improvement*, donde se cambia de solución cuando se encuentra un movimiento que mejora la calidad de la solución actual.

---

**Algorithm 3:** $SwapLS(S)$

---

1: $S_{best} \leftarrow S$
2: $mejora \leftarrow true$
3: **while** $mejora$ **do**
4: $\quad mejora \leftarrow false$
5: $\quad S' \leftarrow S_{best}$
6: $\quad$ **for** $j \in S_{best}$ **do**
7: $\quad\quad$ **for** $i \in V \setminus S_{best}$ **do**
8: $\quad\quad\quad S'' \leftarrow S' \cup \{i\} \setminus \{j\}$
9: $\quad\quad\quad$ **if** $\mathcal{F}(S'') < \mathcal{F}(S')$ **then**
10: $\quad\quad\quad\quad S' \leftarrow S''$
11: $\quad\quad\quad$ **end if**
12: $\quad\quad$ **end for**
13: $\quad$ **end for**
14: $\quad$ **if** $\mathcal{F}(S') < \mathcal{F}(S_{best})$ **then**
15: $\quad\quad mejora \leftarrow true$
16: $\quad\quad S_{best} \leftarrow S'$
17: $\quad$ **end if**
18: **end while**
19: **return** $S_{best}$

---

**Propuesta final**

El algoritmo de selección de medianas propuesto sigue un esquema GRASP multiarranque. En el Algoritmo 4 se muestra el pseudocódigo del algoritmo completo. El procedimiento comienza con una solución $S_{best}$ que irá almacenando la mejor solución conocida. Posteriormente, en el paso 2 se inicia el bucle característico de los algoritmos multiarranque. En cada una de las iteraciones se construirá una solución a través del método *GRASPBuild* (línea 3) y se mejorará con la búsqueda local *SwapLS* hasta encontrar un mínimo local (línea 4). En el caso de que se encuentre una solución mejor, esta es almacenada. Una vez se alcanza el número de iteraciones del algoritmo, se devuelve la mejor solución identificada.

## 4   Resultados experimentales

En esta sección se describen los resultados experimentales obtenidos al aplicar el algoritmo diseñado para el IpMU. Los experimentos se han ejecutado en una maquina virtual equipada con un porcesador AMD EPYC 7282 16-core con 16GB de RAM. El código se ha desarrollado en Java 21.0.1

**Algorithm 4:** $MAT - GRASP(\alpha, V, iters)$

1: $S_{best} \leftarrow \emptyset$
2: **for** $1$ **to** $iters$ **do**
3:  $S \leftarrow GRASPBuild(\alpha, V)$
4:  $S \leftarrow SwapLS(S)$
5:  **if** $\mathcal{F}(S) < \mathcal{F}(S_{best})$ **then**
6:   $S_{best} \leftarrow S$
7:  **end if**
8: **end for**
9: **return** $S_{best}$

utilizando el framework MORK[1] versión 0.20 [13]. Finalmente se ha utilizado `Gurobi 11.0` para la resolución del problema de relajación de aristas descrito en la Sección 3.

Durante la experimentación se ha utilizado el conjunto de instancias presentadas en el artículo previo [9]. Las instancias se dividen en 2 grupos principales según las características del grafo de costes. El primer grupo denominado "Instancias P" genera el grafo de costes de manera que exista una correlación entre los valores de las aristas con el grafo de tiempo de transporte. Más específicamente, $c_a^2 = c_a^1 + U$, donde $U \sim (1, 1.5)$; mientras que en el segundo grupo, denominado "Instancias R", todos los costes son establecidos de manera aleatoria e independiente.

Para ambos conjuntos, el resto de parámetros se toman de los siguientes conjuntos: $n \in \{20, 40, 60, 80\}$, $m \in [100, 500]$, $p \in \{2, 3, 4, 5\}$ y $B \in \{50, 100\}$. Por cada combinación se han construido un total de 5 instancias. Por último, la demanda de cada nodo es $w_i \in [0, 40]$ y el límite de relajación de las aristas es la mitad de su coste total, es decir, $u_a = 0.5 \cdot c_a^2$. Resumiendo, se generan 480 instancias por conjunto, obteniendo un total de 960 instancias donde el algoritmo matheurístico ha sido ejecutado y comparado con el estado del arte.

### 4.1 Selección de parámetros

Para el desarrollo de la propuesta final se realizó una experimentación preliminar para determinar el valor de dos parámetros del experimento:

- El parámetro $\alpha$, que determina el grado de voracidad de la construcción GRASP. En este caso se probaron 5 valores distintos: $\{0.00, 0.25, 0.50, 0.75, 1.00\}$

- La estrategia de búsqueda local, que puede ser *First Improvement* o *Best Improvement* [12].

Por otro lado, el número de iteraciones del algoritmo ha sido establecido de manera experimental tratando de mantener un equilibrio entre la calidad de las soluciones encontradas y el tiempo de ejecución empleado. En este caso se ha establecido un total de 100 iteraciones para el procedimiento multiarranque.

La combinatoria obtiene, por tanto, un total de $10$ configuraciones distintas que fueron probadas en un subconjunto aleatorio de 200 instancias, que representan el $20.8\%$ de las instancias totales estudiadas en el trabajo. Este subconjunto fue obtenido con el propósito de evitar un sobreajuste de los resultados en el conjunto total de instancias[14].

La Tabla 1 muestra los resultados de las 10 configuraciones de parámetros en las 200 instancias de prueba. La primera columna (*Configuración*) muestra la configuración de parámetros estudiada, la segunda columna (*Avg. F(S)*) muestra el promedio del valor de la función objetivo, la tercera columna (*Tiempo(s)*) muestra el promedio del tiempo de ejecución del algoritmo, la cuarta columna (*Dev (%)*) muestra el porcentaje promedio de la desviación del valor obtenido por la configuración respecto al mejor valor obtenido en cada instancia, y finalmente, la quinta columna (*Bests*) muestra la cantidad de veces que el algoritmo alcanza el mejor valor obtenido en términos de la función objetivo.

Como se puede observar, apenas existen diferencias en términos de tiempo de ejecución entre las distintas propuestas, solo existiendo una pequeña mejora en los valores de $\alpha$ menores en la estrategia

---

[1]`https://github.com/mork-optimization/mork`

Table 1: Resultados de las 10 configuraciones del algoritmo en un subconjunto aleatorio de 200 instancias.

| Configuración | Avg. F(S) | Tiempo (s) | Dev. (%) | Bests |
|---|---|---|---|---|
| *Best Improvement*, $\alpha = 0.00$ | 56862.072 | 28.0 | 2.70% | 168 |
| *Best Improvement*, $\alpha = 0.25$ | **55367.159** | 27.8 | **0.00%** | **200** |
| *Best Improvement*, $\alpha = 0.50$ | **55367.159** | **27.6** | **0.00%** | **200** |
| *Best Improvement*, $\alpha = 0.75$ | 55987.270 | 27.7 | 1.12% | 187 |
| *Best Improvement*, $\alpha = 1.00$ | 55926.367 | 27.7 | 1.01% | 188 |
| *First Improvement*, $\alpha = 0.00$ | 56994.953 | 25.4 | 2.94% | 165 |
| *First Improvement*, $\alpha = 0.25$ | 56164.446 | 26.0 | 1.44% | 185 |
| *First Improvement*, $\alpha = 0.50$ | 55610.774 | 27.7 | 0.44% | 195 |
| *First Improvement*, $\alpha = 0.75$ | 56491.112 | 27.9 | 2.03% | 174 |
| *First Improvement*, $\alpha = 1.00$ | 56491.112 | 27.9 | 2.03% | 174 |

*First Improvement*. Sin embargo, la calidad obtenida con estas configuraciones es bastante inferior al resto.

Al analizar las métricas de calidad se observa que los mejores valores son obtenidos por las configuraciones con estrategia *Best Improvement*, que siempre superan a su análogo respecto al parámetro $\alpha$. Entre ellas destacan los valores de 0.25 y 0.50, que siempre alcanzan el mejor valor conocido en todo el conjunto de instancias. Aunque la diferencia de tiempos es ínfima, la configuración seleccionada como propuesta final es *Best Improvement* con $\alpha = 0.50$ debido a su menor tiempo de ejecución frente a $\alpha = 0.25$.

## 4.2 Comparativa con el estado del arte.

La propuesta matheurística basada en GRASP, denominada MAT-GRASP, con la parametrización obtenida en la sección 4.1, se compara en este apartado con el estado del arte conformado por el modelo *FL1* del trabajo [9]. Véase que, aunque este trabajo muestra dos modelos adicionales (*FL2* y *FL3*), los tres certifican la optimalidad de los resultados y el primero es objetivamente superior en términos de tiempo de ejecución.

La comparativa se muestra dividida en dos tablas de resultados que siguen el mismo esquema que la Tabla 1. En la Tabla 2 se muestran los resultados obtenidos en la ejecución del conjunto denominado *Instancias P*, es decir, donde existe una correlación entre los valores del tiempo de transporte y su coste. La Tabla 3 muestra los resultados del conjunto de *Instancias R*, es decir, donde no existe correlación entre ambos valores. Cabe mencionar que, dado que el Modelo *FL1* certifica la optimalidad de sus resultados, la cantidad de mejores resultados obtenidos coincide con la cantidad de soluciones óptimas que los algoritmos alcanzan.

Table 2: Comparativa de resultados en el conjunto de *Instancias P*

| Algoritmo | Avg. F(S) | Tiempo (s) | Dev. (%) | Bests |
|---|---|---|---|---|
| MAT-GRASP | **41457.448** | 27.8 | **0.00%** | **480** |
| Modelo *FL1* | **41457.448** | **4.1** | **0.00%** | **480** |

Table 3: Comparativa de resultados en el conjunto de *Instancias R*

| Algoritmo | Avg. F(S) | Tiempo (s) | Dev. (%) | Bests |
|---|---|---|---|---|
| MAT-GRASP | 69277.884 | **28.1** | 0.00% | 479 |
| Modelo *FL1* | **69276.869** | 42.5 | **0.00%** | **480** |

En la Tabla 2 se observa que el algoritmo matheurístico obtiene la solución óptima en esas 480 instancias, obteniendo las mismas métricas de calidad en términos de valores promedio, desviación y

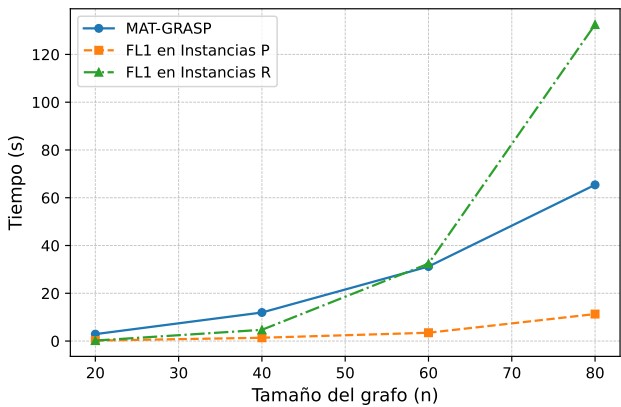

Figure 3: Comparativa de tiempos del algoritmo en comparación al tamaño de la entrada.

mejores resultados alcanzados que el modelo matemático. Sin embargo, se observa que el algoritmo tarda más que el modelo en obtener estos resultados, de 4.1 segundos a 27.8 segundos.

En la Tabla 3 se observa que en el caso del conjunto *Instancias R*, de nuevo el algoritmo propuesto obtiene resultados de muy alta calidad, obteniendo el resultado óptimo en 479 instancias del total de 480. Cabe destacar que el resultado en la instancia que no llega al óptimo se queda a un $0.51\%$ de su valor. Además, en comparación con el conjunto de *Instancias P*, el tiempo de ejecución de MAT-GRASP se mantiene prácticamente invariante, obteniendo un promedio de 28.1 segundos, mientras que en el caso del modelo, cuando se elimina la característica de correlación entre los valores de las aristas, el tiempo de ejecución asciende a 42.5 segundos, por encima de la propuesta de este trabajo.

La Figura 3 muestra un análisis de los tiempos de ejecución del algoritmo en comparación con el tamaño del problema, es decir, el número de nodos del grafo. Como se puede observar, el modelo previo varía su comportamiento según las características de la instancia, mientras que para la propuesta matheurística la dependencia de ambas variables es irrelevante. En el caso de las instancias que presentan correlación entre el tiempo y el coste de transporte el modelo es más rápido que la propuesta de este trabajo, mientras que en el caso contrario el modelo muestra un comportamiento exponencial que supera por mucho el tiempo del algoritmo propuesto.

# 5 Conclusiones y Trabajo Futuro

El IpMU es una variante del problema de la $p$-mediana que generaliza los conceptos de tiempo y coste de transporte. En este trabajo se ha propuesto un algoritmo matheurístico que une una propuesta metaheurística basada en el algoritmo GRASP con un proceso de resolución exacta.

El algoritmo se ha comparado con la resolución del mejor modelo del estado del arte en los dos conjuntos de instancias del trabajo previo, en un total de 960 instancias. En el caso del conjunto de instancias donde las variables son dependientes, a igualdad de calidad, el modelo previo consigue mejores tiempos de ejecución que el algoritmo propuesto. Sin embargo, en el caso de que estas variables sean independientes se verifica el comportamiento opuesto, obteniendo una mejora sustancial en tiempo con el algoritmo propuesto.

Actualmente se sigue trabajando en el desarrollo de la propuesta para este problema. En primer lugar, se está realizando un análisis profundo del subproblema de relajación de aristas, de manera que no se necesiten utilizar herramientas de resolución exacta, previendo una mejora en el tiempo de ejecución del algoritmo.

Por otro lado, se está realizando un análisis del espacio de búsqueda de las instancias en relación a la dependencia de las variables de tiempo y coste, de forma que se pueda ajustar la construcción de soluciones y el tiempo de cómputo, aumentando la eficiencia del algoritmo.

Además se tiene pendiente un ajuste de parámetros más robusto, donde se utilicen herramientas automáticas basadas en test estadísticos, como `irace` [15].

## Agradecimientos y declaración de financiación

Este trabajo forma parte de los proyectos RED2022-134480-T y PID2021-125709OA-C22 financiados por el Ministerio de Ciencia e Innovación (MCIN/AEI/10.13039/501100011033) y FEDER, *una manera de hacer Europa*, así como del proyecto TEC-2024/COM-404 financiado por la Comunidad Autónoma de Madrid.

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
