# OpenReview forum: "Un enfoque matheurístico para el problema de la p-mediana inducida con mejora"
_MAEB/2025/Congreso — MAEB 2025_

### Official Review · Reviewer_ETDi · 2025-03-17
**GRAP para el problema FLP.**

**Rating:** 5
**Confidence:** 3

**Review:**

El artículo aborda problemas del tipo FLP, ubicación de instalaciones.  En su versión sobre grafos, el problema de la p-mediana trata de encontrar p nodos que minimicen la distancia del resto al conjunto de medianas.  Plantean utilizar tiempo de transporte y coste como dos valores en las aristas del grafo.

Utilizan método metaheurístico basado en GRASP, con dos fases, primero selección de medianas, y segunda fase de relajación óptima del grafo.

El artículo está bien escrito, presentar una metodología adecuada y describe resultados de interés, mostrando como la propuesta mejora tiempo de ejecución al comparar con el estado del arte, cuando se trabaja con variables independientes.  Dado que esta parece ser la mejora más destacable, convendría ahondar en este comportamiento, e incluso plantear si existen posibilidades de reducir aún más el tiempo de ejecución, considerando modelos paralelos/distribuidos.

---

### Official Review · Reviewer_3zUZ · 2025-03-17
**Un enfoque matheurístico para el problema de la p-mediana inducida con mejora**

**Rating:** 5
**Confidence:** 5

**Review:**

En este trabajo se resuelve el problema de la p-mediana inducida con mejora. presenta un enfoque matheurístico innovador para el problema de la p-mediana inducida con mejora, combinando modelos matemáticos con la metodología GRASP. Los resultados experimentales son prometedores y están bien documentados, mostrando una mejora significativa en comparación con el estado del arte. Además, el artículo está bien estructurado, con secciones claramente definidas que facilitan la comprensión del problema, la metodología y los resultados.

A continuación, se incluyen algunos comentarios menores:
- La resolución del subproblema de relajación de aristas depende de herramientas de resolución exacta, lo que podría limitar la aplicabilidad del algoritmo en instancias más grandes o complejas. ¿Sabéis decir a partir de qué tamaño del grafo y que número de medianas aparece esta limitación?
- ¿Habéis comprobado el impacto que tiene la calidad de la solución aportada por el GRASP al exacto? Entiendo que al comenzar con una solución cercana al óptimo, la resolución exacta puede enfocarse en un espacio de búsqueda más reducido, lo que mejora la eficiencia del proceso.
- En la Sección Selección de parámetros de parámetros sería interesante ver qué ocurre considerando α uniforme y α random. Es decir, si ejecuta 100 veces, α uniforme serían diferentes valores de α entre [0,1] dando un salto de 1/100, por ejemplo. De forma similar, α random considera que en cada ejecución del algoritmo se toma un α seleccionado aleatoriamente en el intervalo [0,1].

---

### Official Review · Reviewer_z8or · 2025-03-17
**An interesting matheuristic solution to a location facility problem.**

**Rating:** 5
**Confidence:** 4

**Review:**

El artículo propone un algoritmo mateheurístico para abordar el problema de la p-mediana inducida con mejora. La idea es desacoplar la solución del problema en dos fases: problema de las medianas y problema de la mejora.

El artículo es apropiado para MAEB y presenta una serie de puntos a valorar positivamente:

- Se hace una propuesta de relajación de aristas que permite transformar el problema inicial en un problema lineal, reuciendo así la complejidad del modelo inicial.

- El método metaheurístico (GRASP constructivo apoyado en el modelo matemático + búsqueda local) tiene entonces un espacio combinatorio bastante más reducido que abordar.

- La experimentación es extensa tanto en número de instancias como en algoritmos estado-del-arte considerados. Además se hace un grid-search sobre el parámetro que rige la voracidad del método constructivo y la estrategia de búsqueda local (mejor movimiento o primer mejor movimiento).

- El método es competitivo con el estado del arte en exactitud y requiere un 33% menos de tiempo.


No se observan puntos débiles excepto la posibilidad de comparar con otros métodos metaheurísticos, si bien se entiende que esto forma parte de trabajos futuros.

---

### Decision · Program_Chairs · 2025-03-20

Accept